# The Relationship Between Modeling, Caregiver Education, and Diverse Diet in Costa Rican Preschool Children

**DOI:** 10.3390/nu17132087

**Published:** 2025-06-24

**Authors:** Gloriana Rodríguez-Arauz, Benjamín Reyes-Fernández, Georgina Gómez-Salas

**Affiliations:** 1Instituto de Investigaciones Psicológicas, Universidad de Costa Rica, San José 11501-2060, Costa Rica; benjamin.reyesfernandez@ucr.ac.cr; 2Escuela de Medicina, Universidad de Costa Rica, San José 11501-2060, Costa Rica; georgina.gomez@ucr.ac.cr

**Keywords:** maternal education, food parenting practices, modeling, diet diversity

## Abstract

**Background/Objectives:** Previous literature shows that the demographic characteristics of caregivers, such as educational level, determine health processes in young children. However, the mechanisms through which educational level increases health in this population have scarcely been explored in low- and middle-income countries (LMICs). Evidence also suggests that caregivers who model healthy eating to their children, such as eating healthy foods in front of them or enjoying the consumption of healthy foods, increase the likelihood that their child will do the same. **Methods:** Eighty-two primary caregivers of children between 3 and 5 years of age in Costa Rica completed an electronic questionnaire with measures on the highest educational level attained, modeling the healthy eating and diet diversity of their children. Diet diversity indicates the intake of seven important groups of macronutrients that are needed for healthy development in Costa Rican children. **Results:** To investigate these relationships, a simple mediation analysis was carried out, with diet diversity as an outcome, caregiver educational level as a predictor, and modeling as a mediator. The indirect effect of educational level on diet diversity was found to be statistically significant [Effect size = 0.10, 95% C.I. (0.01–0.20)]; **Conclusions:** Results indicate that high educational level is associated with increased usage of healthy eating modeling with children, and this in turn is positively related to their diet being more diverse. Results suggest that the modeling of healthy eating could be the object of interventions aimed at preventing obesity in Costa Rican children.

## 1. Introduction

Childhood obesity remains a major health problem worldwide. According to the World Health Organization [1], an estimated 38.2 million children under the age of 5 had overweight and obesity in 2022. It is recognized that obesity and overweight in children is not only confined to high-income countries; low- and middle-income countries (LMIC) now face a “double burden of malnutrition”. Malnutrition and infectious diseases, which have been present in these settings before, now coexist with obesity in children, contributing to a rise in associated chronic conditions [2].

In Costa Rica, classified as an LMIC, a report from the Health Ministry revealed that 34% of children between 6 and 12 years old were either overweight or obese [3]. This scenario underscores the need to conduct research that identifies key constructs to guide the development of interventions aimed at reducing childhood obesity.

### 1.1. Maternal Education and Obesity Risk in Children

When the health of children is examined, it is necessary to look at the social and environmental conditions in which they develop. One of these conditions is the general health of the primary caregivers, as well as their sociodemographic characteristics. Moreover, this idea is in accordance with the social determinants of health approach [4], in which non-medical factors such as income and education significantly predict health outcomes.

Maternal education stands out as an aspect that is strongly linked to the present and future health status of their children. As mothers increase their educational level, their occupational class and income tend to rise as well, and this has proven to be beneficial for the health outcomes of their offspring. For example, a study [5] found that higher levels of maternal education were associated with behaviors that tended to invest more in their children’s well-being, from 9 months to 5 years of age, across 5000 participants in the United States (US).

In the case of obesity, previous evidence shows that maternal education can play an important role in predicting a child’s risk of being overweight or obese [6,7]. For example, [8] a study with 1900 Iranian children clustered them by socioeconomic and health variables in different risk levels for obesity. Results showed that low educational levels for the main caregiver were the second most significant factor for placing children in the cluster of higher risk for obesity. In England, a similar study with 3717 children conducted a cluster analysis and identified different groups according to their risk for childhood obesity. This study found that children in the lower maternal education and highest deprivation decile showed the highest risk for obesity [9].

Even though the link has been established between higher caregiver educational levels and a lower risk of obesity, it is important to establish the mechanisms through which this relationship finds its strength. One mechanism relates to increased nutrition literacy. Higher educational levels enable a caregiver to be more informed about current dietary guidelines and how healthy or unhealthy certain foods can be. Higher educational levels also relate to increased incomes [10]. It has been reported that people from higher socioeconomic backgrounds tend to meet established dietary guidelines more frequently than those from lower socioeconomic backgrounds [6]. A study with a community sample of 639 Canadian adults found that those of higher incomes were more able to estimate correctly the number of calories contained in a product when reading the nutritional label [11]. Specifically, in Costa Rica, a study conducted with 128 women with 1–4 live births concluded that those with a higher socioeconomic status (SES) reported eating less healthy foods like processed animal-source foods, high-fat condiments, sweetened beverages, and packaged chips, compared to those with those of low SES [12].

Although educational levels can be linked to increased nutrition literacy, other more specific psychological mechanisms can exert an influence, and they have to do with caregivers modeling healthy eating to their children.

### 1.2. Maternal Education and Usage of Modeling

When parents act, their behavior is set as a standard for children to follow. The usage of models as sources of learning in humans has been long studied by psychologists, and its impact on the development of children’s habits is well recognized [13]. However, the realization that modeling can impact a child’s behavior can vary by SES and educational level. There is evidence to support the idea that parents of lower SES tend to model healthy behaviors less frequently to their children [14].

For example, a study with 5729 children and 5183 parents across seven European countries reported that, as parental education increased, parents’ self-reported time spent watching TV tended to decrease [15]. Another study [16] examined the trajectories of physical activity in 408 adolescents in the US across seven assessments. They identified that those adolescents who tended to report high levels of physical activity tended to have higher-SES households, in which both the mother and the father engaged in regular physical activity. A systematic review of 28 studies found that the usage of parental modeling was associated with higher socioeconomic positions for those parents [17].

In conclusion, there is evidence to support the notion that higher SES, which usually relates to higher educational levels, is associated with the usage of the modeling of healthy behavior by parents to their children. This has been noted with different health behaviors, including healthy eating.

### 1.3. Usage of Healthy Eating Modeling and Healthy Eating in Children

Caregivers influence their children’s healthy eating through the example they set for them, with the usage of modeling. Healthy food modeling relates to how caregivers behave around food that is healthy. For example, they can eat it in front of their children, they can also show enjoyment of such foods, and they can choose to eat healthier in front of their children, as opposed to eating unhealthy [6]. A caregiver who models healthy eating sends a powerful message to children that can increase their disposition to try healthy foods and develop a taste for them.

Previous evidence suggests that the modeling of healthy eating positively influences the consumption of nutritious foods [18]. A study of 2.5-to-7-year-old Flemish children and their caregivers found that children’s consumption of unhealthy foods like sugar-sweetened beverages was attenuated if the caregiver refrained from consuming them [19]. Another study conducted with 60 Australian mothers with children 1 year of age found that caregiver modeling predicted the children’s own intake of healthy food when they were 2 years old [20]. Another study with 2382 parent–adolescent dyads in the US [21], found that parental modeling was negatively related to adolescents’ soda consumption.

A study with 1657 US adolescents reported that high levels of perceived parental modeling were related to a higher intake of fruits and vegetables [22]. Similarly, a study with 71 caregivers from the northern Colorado area in the US reported on their fruit and vegetable intake, along with 24 h recalls of the food consumed at home with their children [23]. They found that parental fruit and vegetable intake predicted total fruit and whole fruit intake in the children. A review of 83 studies [24] concluded that there is a strong positive association between the parental intake of healthy foods and the intake of such foods by their children.

In sum, previous evidence shows that, when caregivers act as models of healthy eating, their actions are related to an increased intake of healthy foods by their children.

The question that remains to be asked is: What is the interplay between maternal education, the usage of the modeling of healthy eating by caregivers, and diverse diets in a sample of preschoolers from an LMIC country such as Costa Rica?

### 1.4. This Study

The main goal of this paper is to explore the role of caregiver modeling as a mediator in the relationship between maternal education and diet diversity in Costa Rican children. These relationships are to be explored within the context of Costa Rica, a Central American country that has experienced increased rates of obesity in the past decades, both among adults and children [25]. The food environment in the country is also mixed; healthy foods coexist with highly processed foods as well, and eating healthily is generally considered expensive. A report from the Food and Agriculture indicated that, on average in the world, a healthy meal costs $3.96, while in Costa Rica, the cost is $4.56 [26].

As previously reported, past evidence points out that higher educational levels of main caregivers are linked to improved health outcomes in children, specifically in the intake of healthy food. It is also possible that higher maternal education is related to the usage of modeling, in which caregivers set the example of desirable eating behavior. We also know that the modeling of healthy eating is related to improved consumption of fruits and vegetables and lower consumption of unhealthy foods. Moreover, the evidence previously reported has been collected from high-income countries, so it is important to know whether these relationships are the same in the context of LMICs.

Hence, the study presents the following hypotheses:

**H1.** *Higher maternal education is related to the increased usage of the modeling of healthy eating in Costa Rica*.

**H2.** *The higher usage of modeling is related to a more diverse diet among Costa Rican preschoolers*.

**H3.** *The relationship between maternal education and diverse diet in children is mediated via the usage of modeling by these caregivers*.

## 2. Materials and Methods

### 2.1. Participants and Setting

Eighty-two main caregivers of Costa Rican preschoolers, ages 3 to 5 years, answered an online questionnaire that contained the measures of interest (See database in Appendix A). The inclusion criteria for participants comprised that their children had to be 3 to 5 years old at the moment of answering the questionnaire and that they had no history of developmental disorders. Caregivers also were required to reside in the main urban areas of the country. Urban areas in Costa Rica usually concentrate higher-quality state and social services, access to clean water, electricity, high-speed Internet, and schools; this context lays the ground for a more homogeneous background for all participants. The education distribution of the sample was 4.8% with incomplete secondary education, 8.2% with technical education, 13% with some college education, 50% with complete college education, and 23% with postgraduate education.

Data were collected between August and September of 2020. The country was adjusting to the lockdown imposed via the COVID-19 pandemic, so a question was introduced in the questionnaire to evaluate whether the quality of the participants’ feeding habits had improved, stayed the same, or declined during the months before data collection. The nature of the data is cross-sectional, and the intended analyses were correlational in nature.

### 2.2. Procedure

Caregivers were invited to participate in a larger study on parenting and child nutrition through paid announcements posted on the social media accounts of the Institute of Psychological Research and the School of Psychology. The announcement contained a link to an online questionnaire with measures of interest. Recruitment materials were also sent to groups of parents through WhatsApp (Version 2.20.197.18). When participants clicked on the link for the online questionnaire, a consent form appeared in which they indicated their agreement to participate in the study before giving their answers. All questionnaires and material recruitments, as well as the consent form, were approved by the Ethics Committee of the University of Costa Rica under protocol number CEC-101-2020. Caregivers received no compensation for their participation.

### 2.3. Measures

Parental modeling of healthy eating: This measure was a subscale of the broader Comprehensive Feeding Practices Questionnaire [27], which includes 49 Likert-scale items that assess the frequency of different caregivers’ actions related to feeding practices. The modeling subscale has 4 items: (a) “I try to eat healthy foods in front of my child, even if they are not my favorites”, (b) “I model healthy eating to my child by eating healthy foods myself”, (c) “I try to show enthusiasm towards eating healthy food”, and (d) “I try to show my child how much I enjoy eating healthy foods”. The response format is of 5 points (1 = never to 5 = always). Previous studies have documented a Cronbach’s alpha of 0.80 for this scale. For this study, Cronbach’s alpha was 0.73.Diet diversity: Diet diversity is a simple, valid, and reliable indicator commonly used to signal nutrient adequacy in children [28], hence, it was used as a proxy for healthy eating in this population. This study used a simple food group count to assess diet diversity. Caregivers were asked about the food intake of their children in the past 24 h. Food items reported were categorized into food groups: starchy staples, milk/milk products, eggs, fleshy meat/fish, legumes and nuts, Vitamin A-rich fruits and vegetables, and other fruits and vegetables. This scale ranges from 0 to 7, and scores closest to 7 indicate a more diverse diet.Caregiver’s educational level: The highest out of 8 levels of education completed (e.g., from “some primary school” to “complete college education or more”). Responses were later converted to a numerical scale, ranging from 1 to 8, in which the highest scores indicated the highest levels of education.Body mass index (BMI) of caregiver and child: Caregivers were asked to report on their own height and weight in kilograms and centimeters to calculate their BMI. BMI calculation followed the formula given by the World Health Organization: dividing weight in kilos by height in meters squared. In the case of the target children, this initial BMI percentile score was obtained for each child. A healthy BMI percentile score in children is considered to fall between the ranges of percentile 5 and 85, according to the guidelines by the Centers for Disease Control and Prevention [29].Sex of the child: caregivers were asked to report on the sex of their child between the options of “Male”, “Female”, and “I rather not say”.Income: Access to a list of 13 goods was used as a proxy for this variable. The 13 goods comprise a cellular phone, a landline, a refrigerator, hot water, a water tank, a laptop, a desktop, tablets, a radio or sound system, a car, a motorcycle, a plasma or LCD TV, and paid TV services (cable, satellite, or other paid TV services). Participants indicated that they did or did not possess these items at home, and a score was obtained after the total amount of goods was summed up. If they indicated having the 13 items, they were considered to have the highest incomes.

### 2.4. Statistical Analyses

The online questionnaire was initially answered by 122 people. However, upon a closer review of the data, the other 40 responses were excluded from analyses and not considered for the purposes of this study because the measures of interest did not show at least 80% completion or because participants indicated not being from the urban areas of Costa Rica. All analyses were conducted using the R (version 4.5.0) and RStudio (version 2025.05.0+496) programs, using the packages *car*, *psych*, *lm.beta*, and *mediation.* The PROCESS macro for R [30] was specifically used to further test for simple mediation under Model 4 (See Appendix A for RMardown document). The significance thresholds considered were less than 0.05.

Simple Pearson’s correlational analyses and hierarchical regression analyses were carried out initially to investigate how the variables of interest were related. These correlational analyses were carried out under the consideration that the predictors themselves did not show significant multicollinearity among them. This was achieved through the interpretation of variance inflation values [31], which are advisable to be lower than 5 to suggest acceptable levels of multicollinearity.

To investigate the relationship between educational level, modeling, and diet diversity, a simple mediation analysis [30] was carried out. The outcome variable was diet diversity, the predictor variable was caregiver educational level, and modeling was introduced as a mediator (*c*′ path). This simple mediation model is consistent with more contemporary approaches in testing mediation, as noted in Figure 1.

The method for testing mediation used bootstrapping with indirect effects tested with confidence intervals calculated with 5000 bootstrapping samples. Sex of the child was dummy-coded to be introduced in the analyses as a covariate dichotomous variable, and the number of household items as a proxy for income was also included as a covariate.

For effect sizes, the guidelines by Cohen for Pearson correlation are considered [32], in which a small effect size is 0.10 or below, a medium effect size would be around 0.30, and a large effect size to be around 0.50.

## 3. Results

Table 1 summarizes the data for age, gender, and other demographic variables from both caregivers and their children. All caregivers reported living in the metropolitan area of Costa Rica, which encompasses the main city centers. All of them indicated belonging to the main cultural group in Costa Rica, which is the White–Mestizo. Ninety percent of caregivers indicated that the quality of their habits had stayed the same or improved during the COVID-19 pandemic. Data on the target children was reported by the parents. The children’s mean percentile BMI score fell within the ranges for a healthy height-to-weight ratio, according to their age and gender, using the guidelines provided by the World Health Organization [1], which is from the 3rd to the 95th percentiles.

Initially, to test for H1 and H2, Pearson correlational analyses were run in order to test for the relationship between modeling, educational level, and diet diversity (See Table 2). Significant, positive relationships were found for both variables with diet diversity. In addition, Pearson’s correlational analyses were run to evaluate the relationship between caregiver education and the usage of modeling, *r* = 0.29.

To further the exploration of a possible mediating role of modeling in the relationship between educational levels and diet diversity, a series of hierarchical multiple regression analyses were run on diet diversity, using educational level as a control, along with the sex of the child and the BMI of the main caregiver in an initial step. In the second step, parental modeling was introduced as a predictor. Table 3 shows the results of such analyses. It can be noted that caregiver education showed a significant relationship with diet diversity in the first step, but the significance disappeared when modeling was introduced in the following step. It is also important to note that the amount of variance ($R^{2}$) predicted via Model 1 increased significantly in Model 2.

### Mediation Analyses

The indirect effect of educational level on diet diversity was found to be statistically significant [effect size = 0.10, 95% C.I. (0.01–0.20)]. The results indicate that having a high level of education is associated with caregivers tending to use more healthy eating modeling (path *a*) with their children, and this, in turn, is positively related to their diet being more diverse (path *b*) (see Figure 2). The effects of the covariates sex of the child and amount of household items as a proxy for income were found to be non-significant.

## 4. Discussion

This study proposed that the modeling of healthy eating could act as a mediator in the relationship between caregiver education and diet diversity among preschool children in Costa Rica. Our first hypothesis stated that higher caregiver education was related to the increased modeling of healthy eating, and this hypothesis was supported. Our second hypothesis stated that an increased modeling of healthy eating would be related to increased diet diversity, and this hypothesis was also supported. The third hypothesis stated that modeling could act as a mediator in the relationship between caregiver education and children’s diet diversity. This final hypothesis also found support. In addition to the supported hypotheses, it was found that entering modeling into an initial model explains a significant amount of variance in diet diversity. The results from this study show that previous relationships found in the literature are replicated in the Costa Rican context. Therefore, there is support for the interplay of structural characteristics such as educational levels, psychological constructs such as parental modeling, and nutritional outcomes such as diet diversity. No previous study has achieved this in Costa Rica.

The relationship between caregiver education and modeling of healthy eating could be interpreted from existing evidence, which points to the fact that higher education in a caregiver is related to the intake of healthier foods by children. For example, a study with 849 mothers living in the Netherlands, Belgium, and Germany [33] looked at the differences in social class related to the kinds of food rules the mothers imposed at home. The social class status of the mothers was assessed with a combination of occupation, income, and level of education. The study reports that mothers of a higher social class status tended to restrict sweets, soft drinks, chips, and white bread from their children more often than their lower-social-class status counterparts [33]. More recently, a study [34] conducted with 61,378 children across eleven European cities studied the relationship between the socioeconomic positions of their caregivers during pregnancy and exposure to different hazards during their preschool years. Using logistic regression, they looked specifically at the diet domain, and they found that, consistently across all the cohorts of participants, that a low socioeconomic position of the caregiver was related to a lower odds-ratio for the consumption of fish, fruits, and whole grains and a greater odds-ratio for the consumption of biscuits, chips, and sugar-sweetened beverages [34].

Higher education enables caregivers to understand the importance of having healthy food intake themselves and modeling healthy eating to their children in two ways. One is more automatic, in which caregivers, just by eating healthily themselves or having healthy food available in the home, expose their children to their example, and they are more prone to repeat it. In addition, a more controlled way has to do with either the (1) improvement of health literacy, in which higher education provides the framework for processing complex nutrition information [35], or (2) the development of more intentional mechanisms, through which more educated caregivers consciously decide to model healthy eating to their children [18,22,24,36].

Looking more broadly, the modeling of eating is considered to have a robust effect across different settings and human populations. Since eating often takes place in the presence of other people, it is inevitable that we use others as a guide to adapt our food intake to that of our companions, both in type of food and quantity [36]. A systematic review investigated 69 experimental studies that manipulated the food intake of a social referent to test the effects it had on food intake. These studies were conducted from 1974 to 2014 and included around 5800 study participants. Across diverse samples and using diverse methodologies, social modeling has been found to be a major determinant of one’s intake. This has been established, regardless of age, gender, ethnicity, and weight status and in both hungry and satiated individuals [36]. This effect has been established for both the intake of high-energy, palatable foods and healthier, less palatable foods. In the case of the latter, the authors do mention that repeated exposure is needed for the modeling effect to last.

Especially relevant to this study is the fact that social modeling of food intake is enhanced via the desire of individuals to affiliate with the model or perceive themselves to be like the model [36]. This condition aligns with the relationship between preschool children and their caregivers. Caregivers act as models that children usually want to emulate, identify with, and seek approval from. In this sense, the relationship between children and significant others lays the foundation for examples of healthy eating to be executed, perceived, and repeated.

The mediating role that modeling plays in the relationship between the educational level of the caregiver and their children’s diet diversity also found support. Previous studies have looked at the mediating role various parenting practices have in the relationship between social position and educational level and unhealthy food consumption as well. A study with 639 children from 34 Flemish schools [37], examined the mediating role of various parenting practices in the relationship between SES of the caregiver and specifically soft drink consumption. Using mediational analyses, they found that higher-SES parents tended to have children who consumed 0.42 times the number of soft drinks compared to their lower-SES counterparts. This relationship was mediated via three main parenting practices: permissiveness, having the drinks available at home, and serving those drinks with meals. Interestingly, modeling was not found to mediate the relationship. These counterintuitive findings exemplify the importance of exploring what are the third variables that operate in the relationship between structural characteristics and particular health outcomes. The significant role of modeling as a mediator of the relationship between structural characteristics and diet diversity is also supported by the fact that, once it is entered into the model, it explains a significant amount of variance in comparison to not introducing it [32]. A study such as the one presented here has important strengths, which support significant implications that are worth mentioning. First off, structural characteristics such as the income and educational levels of caregivers act as powerful determinants of healthy eating in children. Evidence of the third variables that mediate this relationship coming from LMIC countries is scarce, so results such as the ones presented contribute to growing knowledge on elements that could be the object of interventions that improve people’s lives.

On a related note, the results presented here matter to the development of interventions that could improve diet diversity in children in a country like Costa Rica, where elicitation research is non-existent. Elicitation research is defined as research that can be conducted previously to the development of an intervention with a specific population to identify salient factors that influence the formation of attitudes and performance of behaviors [38]. It is especially important for interventions in LMIC countries to be focused, cost-effective, simple, and measurable over time since resources are limited in these contexts. In the case of the results presented here, caregiver education is a variable that cannot be directly changed in an intervention, but the modeling of healthy eating can be the focus of a short, psychoeducational, single-component intervention whose impact can be measured and evaluated over time. This is still relevant even though the effect size of the mediation model is considered small [32], given that modeling has an important, robust effect on its own [36], and meta-analyses have reported that its effects are significantly related to healthy eating in children [39].

Providing information to caregivers on different strategies like modeling can be part of broader community interventions on health literacy, which ultimately adds to caregiver empowerment and a sense of control over their children’s health. Previous literature supports the idea that caregivers, despite their low-income status, have an intuition about what healthy food intake looks like, aspire for their kids to eat healthier, and have a healthy relationship with food [40]. More recently, a qualitative study with 25 mothers of 3–6-year-olds in Turkey also found that they expressed their desire to be educated on how to discipline and soothe their children when they were upset, instead of using high-calorie foods for these purposes [41].

The results reported here are not exempt from being considered under a set of limitations. Firstly, since the parents provided their data online, we had no control over the location or the circumstances in which the questionnaire was answered. More controlled conditions are desirable if the study is replicated; participants could answer without distractions or from a single computer. Secondly, the nature of this study was cross-sectional and correlational, which did not allow for inferences on cause and effect or longitudinal relationships. Thirdly, sample size estimations were not calculated before data collection for this study. The sample obtained was dependent on the eligibility criteria of the population and how many subjects were able to answer the questionnaire during the lockdown due to the COVID-19 pandemic. The sample was also small and self-selected (only participants who found the survey appealing answered it), so the conclusions explained here are not intended to be generalized to the Costa Rican population. Therefore, the results detailed are limited in terms of statistical power and generalizability to the Costa Rican population, which calls for caution in interpreting the results [42,43].

Lastly, there are number of variables that can be responsible for the significant relationship between the educational level and diet diversity, such as the availability of healthy food in the home, the temperament and personal characteristics of the child, and the mental state of the caregiver, which were not included here.

Despite the limitations described above, this study provides valuable insights into the interplay between caregiver education, parental modeling, and the intake of healthy foods, particularly with a population of an LMIC. Further exploration is also needed to understand how these findings can translate into focused interventions aimed at increasing health literacy among Costa Rican caregivers, which they can apply to the well-being of their children.

## 5. Conclusions

The results detailed here support the interplay of structural characteristics such as educational levels, psychological constructs such as parental modeling, and nutritional outcomes such as diet diversity in a LMIC country. Further research is needed to replicate them and explore how they translate into interventions that promote healthy eating in preschool children.

## Figures and Tables

**Figure 1 nutrients-17-02087-f001:**
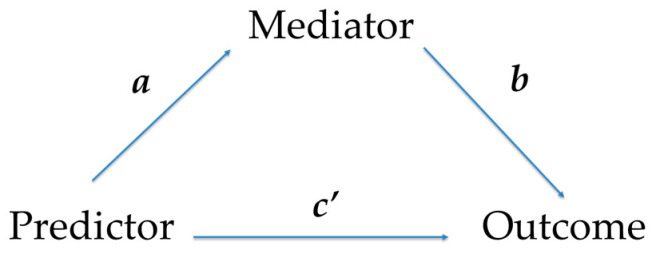
Simple mediation model, where a single mediator is proposed in the relationship between a predictor and an outcome variable. This is consistent with the guidelines proposed for more contemporary approaches in mediation and moderation [30]. Path *a* denotes the relationship between the predictor and the mediator, while path *b* represents the relationship between the mediator and the outcome. *c*′ path refers to the effect of the predictor over the outcome after the mediator is introduced.

**Figure 2 nutrients-17-02087-f002:**
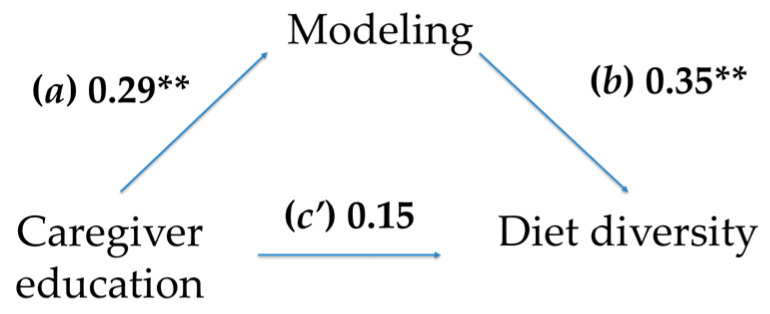
Simple mediation model, where modeling acts as a mediator in the relationship between caregiver education and a child’s diet diversity. Note ** *p* < 0.01.

**Table 1 nutrients-17-02087-t001:** Main demographic characteristics of target children and their caregivers.

	Target Children		Caregivers
	Male	Female	
*n*	40	41	81
Age	M = 4.4 (SD = 0.6)	M = 4.5 (SD = 0.6)	M = 33.2 (SD = 7.2)
Average BMI percentile score	M = 60.4 (SD = 37)	M = 66 (SD = 32)	
BMI			M = 26.5 (SD = 4.6)
Educational level			Some college, complete college, graduate degree: 94%
Access to goods			M = 4.4 (SD = 0.6)

**Table 2 nutrients-17-02087-t002:** Pearson’s correlates between caregivers’ education, modeling, and diet diversity.

Pearson r	Diet Diversity
Caregiver education	0.29 *
Parental modeling	0.39 **

Note: * *p* < 0.05, ** *p* < 0.001.

**Table 3 nutrients-17-02087-t003:** Summary of hierarchical regression analyses to predict diet diversity *n* = 82.

Variable	Model 1			Model 2		
	B	SE B	β	B	SE B	β
BMI of caregiver	0.02	0.02	0.1	0.03	0.02	0.15
Sex of the child	−0.3	0.24	−0.13	−0.24	0.22	−0.11
Caregiver educational level	0.3 *	0.14	0.23	0.17	0.14	0.13
Modeling	-			0.67 **	0.18	0.39
$R^{2}$	0.03			0.16 **		

Note: * *p* < 0.05, ** *p* < 0.001.

## Data Availability

The original contributions presented in this study are included in the article/Appendix A. Further inquiries can be directed to the corresponding author.

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
