# Peer review of "The Relationship Between Modeling, Caregiver Education, and Diverse Diet in Costa Rican Preschool Children"

_nutrients, 2025, doi:10.3390/nu17132087_

Round 1
Reviewer 1 Report
Comments and Suggestions for Authors
Thank you for the opportunity to review this manuscript.
This study explores the role of caregiver modeling as a mediator in the relationship between maternal education and diet diversity among Costa Rican children. The topic is timely and important within the field of nutrition research, and the authors are to be commended for addressing it.
However, frequent grammatical and stylistic issues throughout the manuscript somewhat hindered my ability to engage with the content. A thorough copy-editing pass is recommended. Additionally, improvements to the statistical reporting—particularly in the Methods and Results sections—are necessary to strengthen the rigor and transparency of the research.
Below, I outline my detailed suggestions in the order they appear in the manuscript.
Abstract
The semicolon at the end of the Methods and Results section should be replaced with a period.
The final sentence of the Conclusions section, “Results suggest that modeling of healthy eating could be promoted interventions aimed at preventing obesity in Costa Rican children,” is grammatically incorrect and should be revised for clarity.
Introduction
In the third paragraph, the sentence, “The main goal of this paper is to examine the role of modeling as a mediator in the relationship between maternal education and diet diversity in Costa Rican children,” appears too early. It would be more appropriate to retain this objective where it currently appears in Section 1.4.
Section 1.1
Line 68 requires a comma between “obesity” and “it.”
On line 79, “socioeconomic status” should be spelled out before using the abbreviation (SES), and the period should be placed correctly. The final sentence of this section is unclear in the context of the previous sentence and should be revised for coherence.
Section 1.2
The concluding sentence, “It remains to be seen if parental modeling is related to actual intake of healthy foods by children,” is questionable, given that the next section (1.3) presents evidence supporting the relationship. Consider rephrasing or omitting it.
Section 1.3
The sentence, “A study with Flemish children, ages 2.5 to 7 years old and their caregivers,” is incomplete and requires revision.
On line 121, revise the phrasing to “A study of 1,657 adolescents…” and on line 123, ensure consistent use of the term for the United States (e.g., use either “United States,” “USA,” or “US” consistently throughout). Additionally, remove the question mark at the beginning of the sentence on line 131.
Section 1.4
The first line should read, “The role of caregiver modeling as a mediator,” not “modeling a mediator.” Additionally, please clarify in H2 what is meant by “a more diverse diet.”
Materials and Methods
In Section 2.1, the sentence, “As inclusion criteria for participants included that their children had to be 3 to 5 years old when answering the questionnaire, and no history of developmental disorders,” is awkward and should be rephrased for clarity. It is also unclear why it was necessary for caregivers to reside in urban areas; this criterion should be justified.
In Section 2.2, please identify the “popular text messaging app” used for recruitment. The sentence, “…in which they indicated their agreement with participating in the study before continuing to respond the measures,” is grammatically incorrect and should be revised. Also, be sure to add a period at the end of the paragraph.
In Section 2.3, the first item—“I try to eat healthy foods in front of my child, even they are not my favorites”—is missing a word, likely “if.” In bullet point #4, please spell out the term before abbreviating it (e.g., Body Mass Index (BMI)). In bullet #6, income is defined by possession of 13 goods, but these are not specified—please list them. Also, revise the sentence, “If families possessed 13 out of 13 goods were considered of the highest income,” as it is currently grammatically incorrect.
Section 2.4 lacks discussion of whether multicollinearity was checked, even though variables like maternal education and income are likely to be moderately to strongly correlated. This should be included in the statistical analysis subsection. Effect sizes, which are reported in the Abstract and the Results, should also be introduced here. Furthermore, the manuscript does not address how missing data were handled or the extent of the missing data. Clarify which statistical software was used and specify the threshold p-value used to denote significance. The final sentence, “Sex of the child and amount of household items were included as covariates,” should be clarified—if household items serve as a proxy for income, please state this explicitly. A helpful reference for improving the manuscript’s statistical reporting is: Schaafsma H, Laasanen H, Twynstra J, Seabrook JA. A Review of Statistical Reporting in Dietetics Research (2010–2019): How is a Canadian Journal Doing? Can J Diet Pract Res. 2021 Jun 1;82(2):59–67. doi: 10.3148/cjdpr-2021-005.
Results
In the first paragraph, there is no need to spell out “Body Mass Index” again—use the abbreviation (BMI).
In Table 1, round all means and standard deviations to one decimal place for consistency.
In Table 3, include the sample size in the title to provide clarity about the completeness of data used for regression analysis. Also, ensure the R-squared value is properly squared (R²). Comment on the R-squared value and its interpretation in light of your findings in the Discussion section.
Discussion
Although effect sizes are reported, their interpretation is missing. Please comment on the strength of the reported relationships and their practical significance. It would also be valuable to consider other variables that may influence diet diversity but were not included in the current model. These should be noted as limitations.
In paragraph 2, line 275, the word “out” should be deleted.
In paragraph 3, lines 298–299, the sentence, “As stated in the introduction, example sends a powerful nutritional message to children,” is unclear and should be removed, as it does not add to the discussion.
The paragraph that begins with “The relationship between modeling of healthy eating and diverse diet” appears redundant and reiterates points already made.
On lines 324–325, remove the phrase “these caregivers” at the end of the sentence.
On line 353, replace “can’t” with “cannot.”
On line 367, revise “First off” to “First” or “Firstly.” The sentence that follows—“First off it is worth mentioning that since the parents provided their data online, we were not able to provide a more controlled context in which parents answered the questions”—should be expanded. Why would a more controlled environment have been important? Finally, line 369 marks the first mention that the study is cross-sectional. This should be clearly stated earlier in the Methods section.
One important omission is the lack of discussion around statistical power or sample size justification. Even if this was not a limitation, it should be acknowledged in the Methods section. A helpful resource is: Seabrook JA. How Many Participants Are Needed? Strategies for Calculating Sample Size in Nutrition Research. Can J Diet Pract Res. 2025 Mar 1;86(1):479–483. doi: 10.3148/cjdpr-2024-019.
Author Response
R1.1: This study explores the role of caregiver modeling as a mediator in the relationship between maternal education and diet diversity among Costa Rican children. The topic is timely and important within the field of nutrition research, and the authors are to be commended for addressing it.
Answer: Thank you for your comment.
However, frequent grammatical and stylistic issues throughout the manuscript somewhat hindered my ability to engage with the content. A thorough copy-editing pass is recommended. Additionally, improvements to the statistical reporting—particularly in the Methods and Results sections—are necessary to strengthen the rigor and transparency of the research.
Answer: Thank you for your comment. Your recommendation was taken for this revised version. We are also using the professional editing services provided by the journal. We hope that this revised version is stronger in terms of communication of the results.
Below, I outline my detailed suggestions in the order they appear in the manuscript.
R1.2: Abstract
R1.2.1: The semicolon at the end of the Methods and Results section should be replaced with a period.
Answer: The suggested change was accepted.
R1.2.2: The final sentence of the Conclusions section, “Results suggest that modeling of healthy eating could be promoted interventions aimed at preventing obesity in Costa Rican children,” is grammatically incorrect and should be revised for clarity.
Answer: Thank you for pointing out this mistake. This last sentence in the Conclusions section now reads: “Results suggest that modeling of healthy eating could be the object of interventions aimed at preventing obesity in Costa Rican children.”
R1.3 Introduction
In the third paragraph, the sentence, “The main goal of this paper is to examine the role of modeling as a mediator in the relationship between maternal education and diet diversity in Costa Rican children,” appears too early. It would be more appropriate to retain this objective where it currently appears in Section 1.4.
Answer: Thank you for this suggestion. This paragraph was deleted.
R1.3.1
Section 1.1
Line 68 requires a comma between “obesity” and “it.”
Answer: The suggested change was accepted.
On line 79, “socioeconomic status” should be spelled out before using the abbreviation (SES), and the period should be placed correctly. The final sentence of this section is unclear in the context of the previous sentence and should be revised for coherence.
Answer: Thank you for pointing this out. The whole manuscript was revised so the abbreviation was used after the first time the phrase “socioeconomic status” appeared. I rephrased the last sentence of section 1.4 to read “Although educational level can be linked to increased nutrition literacy, more specific psychological mechanisms can operate, which have to do with caregivers modeling healthy eating to their children.” (Lines 65 to 68 of manuscript with accepted changes).
R1.3.2
Section 1.2
The concluding sentence, “It remains to be seen if parental modeling is related to actual intake of healthy foods by children,” is questionable, given that the next section (1.3) presents evidence supporting the relationship. Consider rephrasing or omitting it.
Answer: Thank you for your comment. The mentioned paragraph was omitted from the revised manuscript.
R1.3.3
Section 1.3
The sentence, “A study with Flemish children, ages 2.5 to 7 years old and their caregivers,” is incomplete and requires revision.
Answer: Thank you for pointing this out. The sentence was revised and merged with the following one, so now it reads: “A study with 2.5 to 7 years-old Flemish children and their caregivers found that children’s consumption of unhealthy foods like sugar-sweetened beverages was attenuated if the caregiver restrained from consuming them” (Line 110 from manuscript with accepted changes).
On line 121, revise the phrasing to “A study of 1,657 adolescents…” and on line 123, ensure consistent use of the term for the United States (e.g., use either “United States,” “USA,” or “US” consistently throughout). Additionally, remove the question mark at the beginning of the sentence on line 131.
Answer: Thank you for pointing this out. The phrasing of this sentence was revised and now it reads “Aa study with 1657 US adolescents in the United States reported that high levels of perceived parental modeling were related to higher intake of fruits and vegetables” (Line 117 of accepted changes manuscript). The term “United States” was searched in all the document and the acronym “US” was used after the first time it appeared. The mentioned question mark was also removed.
R1.3.4
Section 1.4
The first line should read, “The role of caregiver modeling as a mediator,” not “modeling a mediator.” Additionally, please clarify in H2 what is meant by “a more diverse diet.”
Answer: Thank you for your comment. The first line of section 1.4 was corrected as suggested and the phrase “a more diverse diet” in H2 was changed to “increased diet diversity”. What diet diversity means is explained in the materials and methods section.
R1.4 Materials and Methods
R1.4.1: In Section 2.1, the sentence, “As inclusion criteria for participants included that their children had to be 3 to 5 years old when answering the questionnaire, and no history of developmental disorders,” is awkward and should be rephrased for clarity. It is also unclear why it was necessary for caregivers to reside in urban areas; this criterion should be justified.
Answer: Thank you for your comment. The sentence was rephrased to now read: “Inclusion criteria for participants comprised that their children had to be 3 to 5 years old at the moment of answering the questionnaire, and that they had no history of developmental disorders.” We also included a justification on why participants needed to reside in urban areas. This has to do with the fact that in Costa Rica, urban areas are usually more developed and have more access to quality services, so this provided a more homogeneous background for our participants.
R1.4.2: In Section 2.2, please identify the “popular text messaging app” used for recruitment. The sentence, “…in which they indicated their agreement with participating in the study before continuing to respond the measures,” is grammatically incorrect and should be revised. Also, be sure to add a period at the end of the paragraph.
Answer: Thank you for pointing these out. The app was WhatsApp and we included this in the section. We also revised the sentence that you indicated and this one now reads “When participants clicked on the link for the online questionnaire, a consent form appeared, in which they indicated their agreement with participating in the study before giving their answers.” The period was also added at the end of the section.
R1.4.3: In Section 2.3, the first item—“I try to eat healthy foods in front of my child, even they are not my favorites”—is missing a word, likely “if.” In bullet point #4, please spell out the term before abbreviating it (e.g., Body Mass Index (BMI)). In bullet #6, income is defined by possession of 13 goods, but these are not specified—please list them. Also, revise the sentence, “If families possessed 13 out of 13 goods were considered of the highest income,” as it is currently grammatically incorrect.
Answer: The item from the modeling scale in Section 2.3 was completed with the word “if”. The term Body Mass Index was searched throughout the document to make sure that the term was spelled out the first time and the acronym was used after. A list for the 13 items was included and a clarification on how to interpret the goods scale was revised for clarity.
Bullet point #6 now reads:
“Income: access to a list of 13 goods was used as a proxy for this variable. The 13 goods comprise cellular phone, landline, refrigerator, hot water, water tank, lap-top, desktop, tablets, radio or sound system, car, motorcycle, plasma or LCD TV and paid TV services (cable, satellite or other paid TV services). Participants indicated that they possessed or not these items at home and a score was obtained after summing up the total amount of goods. If they indicated having the 13 items, they were considered the highest income (Lines 217-223 of accepted changes manuscript).
R1.4.4: Section 2.4 lacks discussion of whether multicollinearity was checked, even though variables like maternal education and income are likely to be moderately to strongly correlated. This should be included in the statistical analysis subsection. Effect sizes, which are reported in the Abstract and the Results, should also be introduced here. Furthermore, the manuscript does not address how missing data were handled or the extent of the missing data. Clarify which statistical software was used and specify the threshold p-value used to denote significance. The final sentence, “Sex of the child and amount of household items were included as covariates,” should be clarified—if household items serve as a proxy for income, please state this explicitly. A helpful reference for improving the manuscript’s statistical reporting is: Schaafsma H, Laasanen H, Twynstra J, Seabrook JA. A Review of Statistical Reporting in Dietetics Research (2010–2019): How is a Canadian Journal Doing? Can J Diet Pract Res. 2021 Jun 1;82(2):59–67. doi: 10.3148/cjdpr-2021-005.
Answer: Thank you for this comment. All the changes were addressed as follows:
Multicollinearity check: Variance Inflation Values were calculated for the initial hierarchical regression model. These values are 1.06 for caregivers BMI, 1.00 for sex of the child, 1.1 for caregiver education and 1.1 for modeling of healthy eating. According to the reference provided by Neter, J., Wasserman, W. & Kutner, M. H. (1989). Applied Linear Regression Models. Homewood, IL: Irwin. According to their guidelines, values higher than 5 are cause for concern of multicollinearity in regression models. We consider that according to this reference multicollinearity should not be a concern for the results considered here. You are correct regarding the correlation value between the goods scale and caregiver education. This correlation is statistically significant for our sample, and it is 0.25. However, this correlation is to be considered moderate in strength and goods is already considered as a covariate in the model. As we have reported in the analyses section, goods was not a significant covariate.
Effect sizes: a paragraph was added in the Statistical Analyses section (Lines 241-243) and in the Discussion section as well (Lines 383-386)
Missing data: A statement indicating how missing data was handled was added in the Statistical Analyses section in lines 226-228.
Software used and significance thresholds: This was indicated in the Statistical Analyses section in line 229.
Clarification on covariates: See Lines 238-240 in Statistical Analyses section.
R1.5 Results
R1.5.1: In the first paragraph, there is no need to spell out “Body Mass Index” again—use the abbreviation (BMI).
Answer: Thank you for your comment. This was corrected.
R1.5.2: In Table 1, round all means and standard deviations to one decimal place for consistency.
Answer: Thank you for your comment. This was corrected as suggested.
R1.5.3: In Table 3, include the sample size in the title to provide clarity about the completeness of data used for regression analysis. Also, ensure the R-squared value is properly squared (R²). Comment on the R-squared value and its interpretation in light of your findings in the Discussion section.
Answer: Thank you for your comment. Sample size and a corrected R² were added in Table 3. A comment was added in the discussion to point out that introducing modeling as a predictor explains a significant amount of variance as opposed to not introducing it and that this adds to the relevance of modeling as a mediator in the mediation model.
R1.6 Discussion
R1.6.1: Although effect sizes are reported, their interpretation is missing. Please comment on the strength of the reported relationships and their practical significance. It would also be valuable to consider other variables that may influence diet diversity but were not included in the current model. These should be noted as limitations.
Answer: Thank you for pointing this out. A brief elaboration on effect size and the possible influence of unmeasured variables were added in the discussion (Lines 383-386).
R1.6.2: In paragraph 2, line 275, the word “out” should be deleted.
Answer: Thank you for your comment. The word was deleted.
R1.6.3: In paragraph 3, lines 298–299, the sentence, “As stated in the introduction, example sends a powerful nutritional message to children,” is unclear and should be removed, as it does not add to the discussion.
Answer: Thank you for your comment. This sentence was deleted.
R1.6.4: The paragraph that begins with “The relationship between modeling of healthy eating and diverse diet” appears redundant and reiterates points already made.
Answer: Thank you for your suggestion. This paragraph has deleted.
R1.6.5: On lines 324–325, remove the phrase “these caregivers” at the end of the sentence.
Answer: Thank you for your comment. These words were deleted.
R1.6.6: On line 353, replace “can’t” with “cannot.”
Answer: Thank you for your comment. The word was corrected.
R1.6.7: On line 367, revise “First off” to “First” or “Firstly.” The sentence that follows—“First off it is worth mentioning that since the parents provided their data online, we were not able to provide a more controlled context in which parents answered the questions”—should be expanded. Why would a more controlled environment have been important? Finally, line 369 marks the first mention that the study is cross-sectional. This should be clearly stated earlier in the Methods section.
Answer: Thank you for your comment. We revised line 367 and corrected this mistake. We expanded in the discussion on what we mean by controlled conditions and we added a statement in the methods section which indicated that data is cross-sectional.
R1.6.7: One important omission is the lack of discussion around statistical power or sample size justification. Even if this was not a limitation, it should be acknowledged in the Methods section. A helpful resource is: Seabrook JA. How Many Participants Are Needed? Strategies for Calculating Sample Size in Nutrition Research. Can J Diet Pract Res. 2025 Mar 1;86(1):479–483. doi: 10.3148/cjdpr-2024-019.
Answer: Thank you for your comment. This is mainly noted in the limitations section, since no previous power analysis was done in this study. A paragraph was added in the discussion so it is noted that sample size is a significant limitation to the generalizability of results to the Costa Rican population (Lines 403-405 of accepted changes manuscript).
Reviewer 2 Report
Comments and Suggestions for Authors
This paper describes a cross-sectional mediation analysis of the relationship between primary caregiver education level, the modeling of healthy behaviors, and child overweight or obesity in Costa Rica. In general, the literature review, methods, and conclusions are reasonable, with some questions about numbers presented. However, the larger questions are related to:
1) The scientific literature already consistently shows that your hypotheses are true. Is the unique feature that you replicated and demonstrated consistency in Costa Rica? If so, then that needs to be described more explicitly.
2) Many of the examples you provided from the literature were from the US, Australia, and Europe. I also understand that most studies are conducted and published from those locations and that overweight and obesity is still relatively new in LMIC. However, the paper does not describe the context of Costa Rica. Costa Rica is a middle-income country, and I think literacy is quite high there too. I am less sure about the presence of processed foods (probably high) and cost of healthy options (high in the US). I think you need to build out the context more and find articles from other LMIC.
I have some other comments here as well. This will be a combination of detailed edits and broader comments.
- Your introduction is extensive. You can edit it down and it will remain compelling.
- LIne 45: Necessary instead of indispensable
- Line 51 and other locations: you do not need to say "Following this line," You can simply start the sentence
- Lines 55 and 56 - you are focusing on LMIC and use US example and there are potentially different resources. This is one example of my comments above.
- Lines 59-63 - try rephrasing for clarity
- Lines 69-71 - incomplete sentence
- Lines 72-81 - this section feels out of place. You shift to money instead of education. You can reduce this way down. It is well accepted that education is associated with income. Make a statement and cite it.
- Lines 98-103 - same as above
- Line 131 - you can remove the ¿
- Materials and methods: Please provide more information about the distribution of the survey. How was it distributed? What list did you use? What region, etc.
- Provide more detail about your proxy income question line 208. And, you didn't use it in your analysis
- Your sample is very educated. Please provide the breakdown by level of education (the distribution)
- Table 3: I am not sure about the symbols of B and β. What do they represent?
- Figure 1: where did the (b) come from? I don't see that anywhere else?
- I am not sure you need the paragraph in lines 358-367
Author Response
This paper describes a cross-sectional mediation analysis of the relationship between primary caregiver education level, the modeling of healthy behaviors, and child overweight or obesity in Costa Rica. In general, the literature review, methods, and conclusions are reasonable, with some questions about numbers presented. However, the larger questions are related to:
R2.1: 1) The scientific literature already consistently shows that your hypotheses are true. Is the unique feature that you replicated and demonstrated consistency in Costa Rica? If so, then that needs to be described more explicitly.
Answer: Thank you for your comment. It points to the uniqueness of these results, which needs to be described more explicitly. It is correct, scientific literature already provides support for the hypotheses laid out in the manuscript. However, the support for said hypotheses was not provided in the Costa Rican context, and in this regard, no previous study has related structural characteristics such as educational level, psychological constructs (usage of parental modeling) and indicators of diet diversity in children. This was stated more explicitly in the first paragraph of the discussion (lines 300-304 of the accepted changes manuscript)
R2.2: 2) Many of the examples you provided from the literature were from the US, Australia, and Europe. I also understand that most studies are conducted and published from those locations and that overweight and obesity is still relatively new in LMIC. However, the paper does not describe the context of Costa Rica. Costa Rica is a middle-income country, and I think literacy is quite high there too. I am less sure about the presence of processed foods (probably high) and cost of healthy options (high in the US). I think you need to build out the context more and find articles from other LMIC.
Answer: Thank you for bringing this up, since it is such a relevant point. A couple of paragraphs were added that bring some more context about Costa Rica. You are correct, it is considered a LMIC, but also shows a sharp increase in obesity in the past decades as noted by the ELANS study and where eating healthy is considered to be expensive.
R2.3: I have some other comments here as well. This will be a combination of detailed edits and broader comments.
R2.3.1: Your introduction is extensive. You can edit it down and it will remain compelling.
Answer: Thank you for this suggestion. We realize that the introduction is long, but we wish to leave it like that so all the claims that we make remain well supported.
R2.3.2: Line 45: Necessary instead of indispensable
Answer: Thank you for pointing this out. The word “indispensable” was substituted as requested.
R2.3.3: Line 51 and other locations: you do not need to say "Following this line," You can simply start the sentence.
Answer: Thank you for your comment. The phrase was deleted from line 51 and similar instances were located on the manuscript to delete accordingly to reduce wordiness.
R2.3.4: Lines 55 and 56 - you are focusing on LMIC and use US example and there are potentially different resources. This is one example of my comments above.
Answer: Thank you for your comment. This was addressed in the response to comment R2.2
R2.3.5: Lines 59-63 - try rephrasing for clarity
Answer: Thank you for your comment. This was rephrased and now reads “For example, a study with 1900 Iranian children, clustered them by socioeconomic and health variables in different risk levels for obesity. Results showed that low educational level of the main caregiver was the second most significant factor to place children in the cluster of higher risk for obesity.” (Lines 55-58 of revised manuscript)
R2.3.6: Lines 69-71 - incomplete sentence
Answer: Thank you for your comment. The sentence was complete and divided into two separate ideas. These lines now reads: “One mechanism relates with increased health literacy. Higher educational levels enable the caregiver to be more informed about current dietary guidelines and how healthy or unhealthy certain foods can be.” (Lines 65-67 of accepted changes manuscript).
R2.3.7: Lines 72-81 - this section feels out of place. You shift to money instead of education. You can reduce this way down. It is well accepted that education is associated with income. Make a statement and cite it.
R2.3.8: Lines 98-103 - same as above
Answer: Thank you for your comment. A reference was introduced to state clearly how education is associated with income (Lines 67-68) . He hope that after this citation, it becomes clear what further studies have found regarding SES and healthier behavior patterns in adolescents.
R2.3.9: Line 131 - you can remove the ¿
Answer: Thank you for pointing this out. The question mark was deleted.
R2.3.10: Materials and methods: Please provide more information about the distribution of the survey. How was it distributed? What list did you use? What region, etc.
Answer: Thank you for your comment. The survey was not distributed according to a particular participant list, but rather distributed through direct advertising used through Facebook, in which we paid for the announcement to be seen mainly by participants from urban areas from Costa Rica. We also distributed the survey through WhatsApp parent groups and we advertised the survey also through the social media accounts of the School of Psychology and the Institute of Psychological Research.
R2.3.11: Provide more detail about your proxy income question line 208. And, you didn't use it in your analysis.
Answer: Thank you for your comment. In line with a comment from another reviewer, more information was added on the proxy income question, which now reads:
“Income: access to a list of 13 goods was used as a proxy for this variable. The 13 goods comprise cellular phone, landline, refrigerator, hot water, water tank, lap-top, desktop, tablets, radio or sound system, car, motorcycle, plasma or LCD TV and paid TV services (cable, satellite or other paid TV services). If participants indicated that they possessed or not these items at home and a score was obtained after summing up the total amount of goods. If they indicated having the 13 items, they were considered the highest income.”
This variable was used in the analyses as a covariate in the simple mediation model, as stated in Line 240 of accepted changes manuscript.
R2.3.12: Your sample is very educated. Please provide the breakdown by level of education (the distribution)
Answer: Thank you for pointing this out. The distribution of the sample in terms of education is 4.8% has incomplete secondary education, 8.2% have technical education, 13% some college education, 50% complete college education, 23% postgraduate education. This distribution was added in Section 1.2.
R2.3.13: Table 3: I am not sure about the symbols of B and β. What do they represent?
Answer: Thank you for pointing this out. The symbol B indicates the unstandardized regression coefficients product of the hierarchical regression analyses for each one of the predictors introduced in either Model 1 and Model 2. β is the accepted notation for the standardized coefficients. The difference between the two is that the unstandardized coefficients use the raw data and are more difficult to interpret, since the set of parameters have different units. Standardized coefficients make interpretation easier, since all the parameters can be interpreted in terms of standard deviations and this brings uniformity to the interpretation.
R2.3.14: Figure 1: where did the (b) come from? I don't see that anywhere else?
Answer: Thank you for your comment. In Figure 1, the (b) refers to path b, which appears in line 283 of the accepted changes manuscript. This path in the figure represents the relationship between caregiver modeling and diet diversity.
R2.3.15: I am not sure you need the paragraph in lines 358-367
Answer: Thank you for your comment. We consider that the paragraph can be kept in the manuscript if more context is provided on how modeling can be part of broader community interventions that ultimately provide empowerment and a sense of self-control to caregivers. The literature on this matter points out that caregivers express a need for these strategies to be delivered to them so they can help their kids eat healthier, in spite of their low income. We provided some rephrasing so we hope that the paragraph is clearer now. (Lines 387-395 of revised manuscript)
Round 2
Reviewer 1 Report
Comments and Suggestions for Authors
The authors have adequately addressed most of my comments but revisions to the statistical analysis subsection, statistical reporting, and sample size justification are needed.
The first line now reads "The online was initially answered by 122 people." The online what?
The transition from 122 to 82 subjects is somewhat confusing. The phrase "upon closer review of the 226 data" is unclear—does "226" refer to the number of items or cases? This should be reworded for clarity. Additionally, the exclusion of 40 responses due to incompleteness is acceptable, but the criteria for determining incompleteness should be specified (e.g., percentage of missing data per case, key variables missing, etc.).
The mention of using R and RStudio is appropriate. However, the version numbers of R, RStudio, and any key packages used (e.g., lavaan, psych, or mediation) should be reported for reproducibility.
The mediation model is briefly described, but it lacks sufficient detail. It would be helpful for the authors to specify the method used (e.g., Baron and Kenny approach, bootstrapping), and how indirect effects were tested (e.g., confidence intervals, number of bootstrap samples). Clarify whether standardized or unstandardized coefficients are reported. Additionally, the naming of the “c prime path” should be corrected to “c′ path” or "direct effect" for clarity and consistency with mediation terminology.
The authors state "For effect sizes, the guidelines by Cohen are considered [32] in which a small effect size is .10 or below, and medium effect size would be around 0.25 and a large effect size to be around .40." The authors do not clarify which effect size metric these thresholds apply to, which is essential. Either way, the stated values (.10, .25, .40) do not match Cohen’s standard for r, d, or f². I assume these effect sizes were for Pearson's correlation coefficient. Is that accurate? If yes, for effect size interpretation, Cohen’s guidelines for Pearson’s r indicate values of approximately .10, .30, and .50 correspond to small, medium, and large effect sizes, respectively (Cohen, 1988).”
As noted in my initial review, the following two papers would significantly strengthen the manuscript. Citing them would demonstrate adherence to established statistical reporting guidelines and provide important context for the discussion of sample size considerations:
Schaafsma H, Laasanen H, Twynstra J, Seabrook JA. A Review of Statistical Reporting in Dietetics Research (2010-2019): How is a Canadian Journal Doing? Can J Diet Pract Res. 2021 Jun 1;82(2):59-67. doi: 10.3148/cjdpr-2021-005.
Seabrook J.A. PhD. How Many Participants Are Needed? Strategies for Calculating Sample Size in Nutrition Research. Can J Diet Pract Res. 2025 Mar 1;86(1):479-483. doi: 10.3148/cjdpr-2024-019. Epub 2024 Dec 6. Erratum in: Can J Diet Pract Res. 2025 May 13:1. doi: 10.3148/cjdpr-2025-207
Author Response
R1: The authors have adequately addressed most of my comments but revisions to the statistical analysis subsection, statistical reporting, and sample size justification are needed.
Answer: Thank you for your feedback. The specific comments on the statistical analysis subsection, statistical reporting are addressed below. Regarding sample size, a formal a priori sample size calculation was not conducted for this study. However, we believe that the sample reflects important characteristics of the eligible Costa Rican population that was accessed during the study period, under the lockdown from the COVID-19 pandemic. We understand that this is a significant limitation under current guidelines for statistical reporting on sample size provided by Seabrook (2025), so we expanded on this in the limitations section (Lines 417-425 of accepted changes manuscript).
R1: The first line now reads "The online was initially answered by 122 people." The online what?
Answer: We apologize for this lack of clarity. The word that goes after “online” is “questionnaire” and it is now corrected (Line 226 of manuscript with accepted changes)
R2: The transition from 122 to 82 subjects is somewhat confusing. The phrase "upon closer review of the 226 data" is unclear—does "226" refer to the number of items or cases? This should be reworded for clarity. Additionally, the exclusion of 40 responses due to incompleteness is acceptable, but the criteria for determining incompleteness should be specified (e.g., percentage of missing data per case, key variables missing, etc.).
Answer: We apologize for the lack of clarity in this explanation. We believe that the number 226 refers to the line number of the paragraph and has nothing to do with the explanation. However, the paragraph was reworded and expanded to include more details on the reasons on why 40 cases were deleted. The paragraph now reads:
“The online questionnaire was initially answered by 122 people. However, upon closer review of the data, the measures of interest were only complete for 82 subjects. The other 40 responses were excluded from analyses and not considered for the purposes of this study because the measures of interest did not show at least an 80% completion or because participants indicated not being from the urban areas of Costa Rica.”(Lines 226-232 of manuscript with accepted changes)
R4: The mention of using R and RStudio is appropriate. However, the version numbers of R, RStudio, and any key packages used (e.g., lavaan, psych, or mediation) should be reported for reproducibility.
Answer: Thank you for your comment. The paragraph was completed with the requested information and it now reads:
“All analyses were conducted using the R and RStudio programs, using the packages car, psych, lm.beta and mediation. The PROCESS macro for R, proposed by Hayes (2022), was specifically used to further test for simple mediation, under Model 4. (Lines 230-232).
R5: The mediation model is briefly described, but it lacks sufficient detail. It would be helpful for the authors to specify the method used (e.g., Baron and Kenny approach, bootstrapping), and how indirect effects were tested (e.g., confidence intervals, number of bootstrap samples). Clarify whether standardized or unstandardized coefficients are reported. Additionally, the naming of the “c prime path” should be corrected to “c′ path” or "direct effect" for clarity and consistency with mediation terminology.
Answer: Thank you for pointing this out. A figure with the proposed simple mediation model was added to add clarity as to what was expected in mediation and what was obtained with the results. It was specified that bootstrapping method was used to test for mediation and that indirect effects were tested with confidence intervals using 5000 bootstrapping samples. We also corrected with all mentions of c prime, to c′ path (Lines 246-253 of accepted changes manuscript).
R6: The authors state "For effect sizes, the guidelines by Cohen are considered [32] in which a small effect size is .10 or below, and medium effect size would be around 0.25 and a large effect size to be around .40." The authors do not clarify which effect size metric these thresholds apply to, which is essential. Either way, the stated values (.10, .25, .40) do not match Cohen’s standard for r, d, or f². I assume these effect sizes were for Pearson's correlation coefficient. Is that accurate? If yes, for effect size interpretation, Cohen’s guidelines for Pearson’s r indicate values of approximately .10, .30, and .50 correspond to small, medium, and large effect sizes, respectively (Cohen, 1988).”
Answer: Thank you for pointing this out. The effect size guidelines stated in the manuscript correspond indeed to those for Pearson’s correlation by Cohen (1988). The paragraph was corrected to include the metric and the correct values for the guidelines (Lines 256-258 of revised manuscript).
R7: As noted in my initial review, the following two papers would significantly strengthen the manuscript. Citing them would demonstrate adherence to established statistical reporting guidelines and provide important context for the discussion of sample size considerations:
Schaafsma H, Laasanen H, Twynstra J, Seabrook JA. A Review of Statistical Reporting in Dietetics Research (2010-2019): How is a Canadian Journal Doing? Can J Diet Pract Res. 2021 Jun 1;82(2):59-67. doi: 10.3148/cjdpr-2021-005.
Seabrook J.A. PhD. How Many Participants Are Needed? Strategies for Calculating Sample Size in Nutrition Research. Can J Diet Pract Res. 2025 Mar 1;86(1):479-483. doi: 10.3148/cjdpr-2024-019. Epub 2024 Dec 6. Erratum in: Can J Diet Pract Res. 2025 May 13:1. doi: 10.3148/cjdpr-2025-207
Answer: Thank you for providing these valuable references. Both of them were revised and cited in the paper, as support for the fact that the lack of sample size a priori calculation is in fact an important limitation according to recent guidelines in dietetics research.
Reviewer 2 Report
Comments and Suggestions for Authors
The comments were addressed.
Author Response
Thank you for your thoughtful feedback